# Enhancing anomaly detection in distributed power systems using autoencoder-based federated learning

**Kimleang Kea**[ID][1], **Youngsun Han**[1], **Tae-Kyung Kim**[ID][2]*

**1** Department of AI Convergence, Pukyong National University, Nam-gu, Busan, South Korea, **2** Department of Computer & Information Technology, Incheon Jaeneung University, Dong-gu, Incheon, South Korea

* misoh049@jeiu.ac.kr

**Data Availability Statement:** Public dataset: https://archive.ics.uci.edu/ml/datasets/individual +household+electric+power+consumption Public source code: https://doi.org/10.5281/zenodo. 8036661.

## Abstract

The growing use of Internet-of-Things devices in electric power systems has resulted in increased complexity and flexibility, making monitoring power usage critical for effective system maintenance and detecting abnormal behavior. However, traditional anomalous power consumption detection methods struggle to handle the vast amounts of data generated by these devices. While deep learning and machine learning are effective in anomaly detection, they require significant amounts of training data collected on centralized servers. This centralized approach results in high response time delays and data leakage problems. To address these challenges, we propose an Autoencoder-based Federated Learning method that combines the AutoEncoder and Federated Learning networks to develop a high-accuracy algorithm for identifying anomalies of power consumption data in distributed power systems. The proposed method allows for decentralized training of anomaly detection models among IoT devices, reducing response time and eventually solving data leakage issues. Our experimental results demonstrate the effectiveness of the FLAE method in detecting anomalies without needing data transferring.

## 1 Introduction

As technology advances and people find new ways to utilize power in their daily lives, power consumption in households is increasing exponentially. To address this challenge, smart grids are using data collected by Internet-of-Things (IoT) devices to optimize various outcomes such as cost reduction, safety awareness, and prevention of equipment downtime. The integration of smart devices has led to a reduction in the number of required devices and, consequently, a decrease in the volume of corresponding data streams. However, power system companies still struggle to monitor all their devices simultaneously, resulting in unnecessary power wastage. Thus, it is crucial to detect unusual power consumption to avoid unnecessary operational expenses and improve efficiency.

Anomaly detection in power consumption data has become crucial for tenants to identify abnormal consumption patterns, detect malfunctioning electrical appliances, and reduce power consumption costs. It involves identifying anomalous observations that deviate from

**Funding:** This work was conducted under the framework of the research and development program of the Korea Institute of Energy Research (C1-2418). The funders had no role in study design, data collection and analysis, decision to publish, or preparation of the manuscript.

**Competing interests:** The authors have declared that no competing interests exist.

the typical pattern in a given dataset [1]. With the advent of IoT applications, detecting abnormalities in IoT time-series data has gained significant importance [2, 3]. For instance, a light bulb's power consumption pattern varies based on the time of day and the availability of natural light in residences. Thus, high power consumption levels during working hours may be anomalous, while the same during nighttime hours may not be anomalous. Anomalies are classified into three types: point anomalies, contextual anomalies, and collective anomalies. A point anomaly occurs when a data point is excessively high or low compared to others. In this paper, we define abnormal power consumption as the difference between predicted and actual power consumption that exceeds a threshold during a given period, indicating that the current period's power consumption is anomalous. Therefore, the primary objective of this paper is to identify point anomalies in power consumption data.

The early works on prediction and detection of anomalies in power systems have mostly focused on centralized approaches, where data is collected from various sources and analyzed in a centralized location. Wadi et al. [4] employed machine learning techniques to detect faults in power system networks. However, the primary limitation of their approach was that it relied on supervised learning during the training process, which necessitated the use of labeled data containing abnormal instances. Labeled datasets are critical in supervised learning to train machine learning models. However, producing these datasets requires expertise to ensure both the quality and quantity of data. Li et al. [5] proposed a novel approach for centralized water leak detection by utilizing acoustic emission sensor data from municipal pipeline systems. Their method leverages a classifier based on artificial neural networks to identify anomalies in the sensor data, which can indicate the presence of water leaks in the pipeline system. The limitation of thier work is the reliance on a centralized approach. Zhang et al. [6] proposed an approach to detect abnormalities in a power consumption dataset by employing a transformed K-means algorithm. The algorithm clusters the data into groups based on their similarity, and any data points outside of these clusters are identified as anomalies. While the transformed K-means approach is an effective method for anomaly detection, the centralized dataset approach may not be suitable for real-time anomaly detection applications, as the data must be collected, processed, and analyzed in a centralized location. This could potentially increase data transfer bandwidth and expose sensitive information about sensors, leading to privacy breaches.

In a centralized system, abnormalities in devices are detected using a server-based model that gathers training data from each device via gateways and stores it on a centralized server. Hence, centralized servers are often challenging to maintain due to their vulnerability to cyber-attacks, which raises concerns about data privacy. Additionally, transmitting data to a centralized server requires high bandwidth. To address these concerns, decentralized learning solutions have been developed. These solutions enable only locally trained models to be transmitted to a centralized server, thereby allowing all private data to remain local. One such solution is Federated Learning (FL), which is an emerging and robust decentralized approach to training machine learning models collaboratively on local devices, using decentralized datasets [7–12]. This approach enables data to remain at the source of its generation, resulting in reduced data transmission costs [13]. While FL presents significant privacy advantages over centralized data collection, it can be challenging to integrate into IoT systems due to several factors. The primary challenge is device communication, dataset, and model heterogeneity, which can hinder the effectiveness of the FL process. To address heterogeneity issues in IoT environments, the authors in [14] have proposed personalized FL methods to mitigate the negative effects of heterogeneity. However, one significant challenge is the gradient leakage problem, as model gradients must be shared and aggregated on a central server. This presents various cyberattack techniques to intercept gradient data packages. To address this, data leakage can be mitigated through the use of FL with secure multi-party computation,

homomorphic encryption, and differential privacy techniques. As a result, FL plays a critical role in (1) detecting anomalous devices, (2) reporting abnormal devices, and (3) preventing privacy leakage from devices.

In this paper, we propose a power consumption anomaly detection model that integrates the autoencoder (AE) network with the unique features of FL. This approach, which we refer to as the FLAE method, leverages the strengths of both models to improve power consumption anomaly detection. Our research makes the following contributions:

- We adopt the autoencoder which is a type of neural network that is well-suited for anomaly detection. The autoencoder network is trained to reconstruct normal data, making them effective at identifying anomalies that deviate significantly from normal data patterns.

- We introduce a decentralized approach for power consumption anomaly detection that utilizes the combination of federated learning framework with the autoencoder network. Our proposed approach allows IoT devices to train a single anomaly detection model collaboratively using their own datasets locally to identify a wider range of anomalies and improve overall accuracy.

- We implement a dynamic threshold selection strategy that utilizes the peak over threshold algorithm for the automatic selection of an appropriate threshold based on the data distribution, ensuring that anomalies are accurately detected while minimizing false positives.

- We showcase the effectiveness of our proposed framework by conducting experimental comparisons with state-of-the-art models. Moreover, we demonstrate the efficiency of the FLAE method in both homogeneous and heterogeneous configurations of power consumption datasets.

This paper is organized as follows. Section 2 provides background information about anomalies in time-series data and FL. Section 3 provides an overview of work related to anomaly detection in both machine learning and FL. Section 4 describes how the AE and FL are combined in the proposed FLAE method, the problem statement, the data preprocessing techniques, and the proposed anomaly detection architecture. Section 5 discusses the experimental setup and reports the performance of the state-of-the-art models and our proposed FLAE method. Finally, conclusions are presented in Section 6.

## 2 Background

### 2.1 Anomaly in time-series data

Anomalies in time-series data refer to data points that exhibit abnormal behavior and significantly deviate from the expected pattern of the previous timestamp. These anomalies may arise from different factors, including outliers, measurement errors, or sudden changes in the underlying process. They can pose a challenge in data analysis and must be carefully identified and addressed to avoid misleading insights and decisions. Depending on the application, anomalies can be referred to as novelties, deviants, or outliers [15]. In this paper, the term anomaly will be used for consistency. According to [1], there are three main categories of anomalies related to time-series data, i.e., point anomalies, contextual anomalies, and collective anomalies.

1. Point anomalies is occurring when a singular data point is significantly different from the rest of the dataset in terms of its properties or characteristics.

2. Contextual anomalies are a type of anomaly where a data instance is considered anomalous only in a particular context or setting, while it may be entirely normal in other contexts.

3. Collective anomalies refer to a type of anomaly where a group or a set of related data instances display anomalous behavior or characteristics within the context of the entire dataset.

## 2.2 Federated learning

FL is a distributed machine learning technique introduced in [16, 17]. FL enables a model to be trained using local data on decentralized edge devices without the need to exchange this data between devices. This is achieved by allowing each device to perform the necessary training locally. After performing local training, the device transmits its results to a central server. Many devices participate in the training process, contributing model updates to the server. These updates are then aggregated using the Federated Averaging (FedAvg) algorithm, which produces an improved global model.

The FedAvg algorithm [16] works by randomly selecting a subset of devices each round, and calculating the average weight of their local models. This average weight is then used to generate a weighted average of all device model results, which is transmitted to the server. FedAvg is an effective optimization algorithm for training loss and is also robust against non-IID distributions and unbalanced data since each device's data differ and may not follow the same distribution. FL provides several key benefits including the following critical advantages.

- Training time reduction: Parallel calculations of gradients are performed on multiple devices, which reduces calculation times significantly.

- Inference time reduction: Each device has its own copy of the model; thus, predictions can be made extremely quickly without dependence on server queries.

- Privacy reservation: A major privacy risk can occur when sensitive information is uploaded to the server; thus, storing data locally helps end users maintain privacy.

- Ease of collaborative learning: Rather than collecting a single massive dataset to train a model on the server, FL allows the end users to train the model in a local manner.

## 3 Related work

Anomaly detection in time-series data is a complex task, and many studies have investigated power consumption data prediction and abnormality detection because this information is critical in terms of realize effective and efficient power systems.

## 3.1 Machine learning anomaly detection

In the early stages of deep learning, many methods [4, 6, 18–20] are used and produce significant results in detecting abnormalities in data. Wadi et al. [4] used machine learning-based techniques to detect faults in power system networks. The methods used were the principal component analysis and one-class support vector machine techniques. These methods obtained an accuracy of 79.28% to 79.84% and an ROC score of 0.67 to 0.73. However, their methods used supervised learning during training, which heavily depends on the anomalous labels in the dataset. Mao et al. [6] introduced the isolation forest (iForest) algorithm to detect abnormalities in power consumption data. This algorithm is used to detect anomalous data points. The iForest algorithm is used to detect data points that are inconsistent with other data patterns. These data points are then marked as abnormal. However, the iForest algorithm requires a large amount of training data to be able to detect anomalous data points accurately. Additionally, if the algorithm is not optimized correctly, the implementation can take a long

time and require a lot of computer resources. Zhang et al. [20] proposed a method for predicting power consumption and detecting anomalies that combines the Transformer model with the K-means clustering approach. There are two steps to the proposed method: predicting anomalies, and detecting them. A transformer model is used to predict power consumption, and K-means clustering is used to improve the accuracy of the predictions. Anomalies are then detected by comparing predicted and actual values. The majority of related research is constrained by the collection of training data through a centralized server. This approach can result in privacy breaches and increased computational resources required for training. Thus, new approaches are necessary to overcome these limitations.

## 3.2 Federated learning anomaly detection

The FL was initially proposed by Google researchers [16] for tasks such as image classification and next word prediction. With its increasing popularity, it can also be utilized for anomaly detection. Sater et al. [21] proposed FL in a stack LSTM model to train time-series IoT data in smart building. The FL approach has proven to be efficient for demonstrating the performance of models on datasets that resemble the actual data generated by a electric smart building. To train their model, the authors utilized the FedAvg algorithm, repeating the process until the model achieved convergence or the maximum number of training rounds was reached. In the training process, data was collected from multiple heterogeneous devices to tackle the common challenges faced by IoT devices. However, this method demanded significant computational resources and time to enable effective training and real-world applications. Consequently, from an IoT hardware perspective, LSTM models are generally considered inefficient. Ayed et al. [22] found that FL is feasible for network-based intrusion detection. The goal of their study was to distinguish between normal and abnormal network traffic behavior by analyzing network traffic data. They proposed a FL algorithm that is resistant to anomalies caused by malicious adversaries. The algorithm is able to maintain performance levels even when confronted with new and unknown attacks. The proposed approach used a state-of-the-art dataset and simulated its distribution over a set of clients. Their experimental result show that this method is successful in identifying security breaches. Nguyen et al. [23] propose an anomaly detection system that uses FL to detect compromised IoT devices by aggregating anomaly detection profiles. To their extensive evaluation, their proposed system is highly effective (95.6% detection rate) and fast at detecting devices compromised by the infamous Mirai malware.

## 4 Methodology

In this section, we delve into the time-series problem and explore preprocessing techniques for time-series data. We then introduce the FL architecture which utilizes the AE network to address time-series challenges. We also examine anomaly detection and its threshold selection strategy, allowing for a comprehensive understanding of the method.

### 4.1 Problem statement

A time-series contains a sequence of data points collected at equally spaced timestamps involving multiple variables [24]. In this study, we only focused on a multivariate time-series defined as follows:

$$\mathcal{T} = \{x_1, \ldots, x_T\} \tag{1}$$

where $T$ is the length of $\mathcal{T}$, and each data point $x_t$ is collected at a specific time $t$. An input of

multivariate time-series is denoted $x \in \mathbb{R}^{n \cdot k}$, where $n$ is the maximum timestamp length, and $k$ is the number of input features. In the anomaly detection task, given the time-series input $\mathcal{T}$ to identify an unseen observation $\hat{\mathcal{T}}$, we must predict $y = \{y_1, \ldots, y\hat{T}\}$ where $y_t \in \{0, 1\}$ indicates whether the data point at $t^{th}$ timestamp is an anomaly. The number of the unseen observation $\hat{\mathcal{T}}$ and the normal set of test data $\mathcal{T}$ differ is measured using an anomaly score, which is compared to a threshold to obtain a corresponding anomaly label.

## 4.2 Data preprocessing

In this study, we applied two data preprocessing techniques, i.e., noise reduction and normalization, and these techniques are described in detail in the following.

**4.2.1 Noise reduction.** We used the Savitsky-Golay filter to reduce noise in the raw sequence before inputting it to the anomaly detection models [25]. This filter uses least-squares polynomial fitting, which is a low-pass filter that replaces each value in the series with a new value obtained from a polynomial fit. The general expression for the filter is as follows:

$$g_i = \sum_{i=-m}^{+m} c_i \cdot \frac{f_{i+1}}{N} \tag{2}$$

where $f_i$ is the original time-series data, $g_i$ is the smoothed value which is a linear combination of $c_i$ and $f_i$, $c_i$ is given as the polynomial of certain degree maintains higher values, and $N$ is the convoluting integer which is equal to the smoothing window size comprising (2m+1) points [26].

**4.2.2 Normalization.** We normalized the time-series data using the min-max transformation formula to make the model more robust. All of the features will be transformed in the range [0, 1]. This scaling will distort the original distribution for the dataset. To model the dependence between data point $x_t$ at the current timestamp and the previous ones, we define a window of length $K$ at time $t$ as follows.

$$W_t = \{x_{t-K+1}, \ldots, x_t\} \tag{3}$$

We convert time-series $\mathcal{T}$ into a sequence of window $W = \{W_1, \ldots, W_T\}$ to be used as the training input and $\hat{W}$ as the test time-series data. Here, rather than predicting the anomaly label $y_t$ for each input window $W_t$ directly, we need to calculate the anomaly score $s_t$ for this window [27]. The goal of the anomaly detection problem is to use anomaly scores $s_t$ for the previous windows to calculate the threshold value $D$. The input window is labeled as anomalous if $y_t = 1(s_t \geq D)$. The anomaly score is the measurement of how different the original input window is from the reconstructed input window. It is calculated by taking the deviation between the two.

## 4.3 AutoEncoder model

An AE is an unsupervised learning technique proposed by Hinton and Zamel [28] that comprises an encoder $E$ and decoder $D$. Here, the encoder maps input $X$ to a set of latent variables $Z$. In contrast, the decoder maps the latent variables $Z$ back into the input space as a reconstruction $R$, as shown in Fig 1. The deviation between the original input $X$ and reconstruction $R$ is referred to as the reconstruction error, which is defined as follows:

$$L_{AE} = \|X - AE(X)\|_2 \tag{4}$$

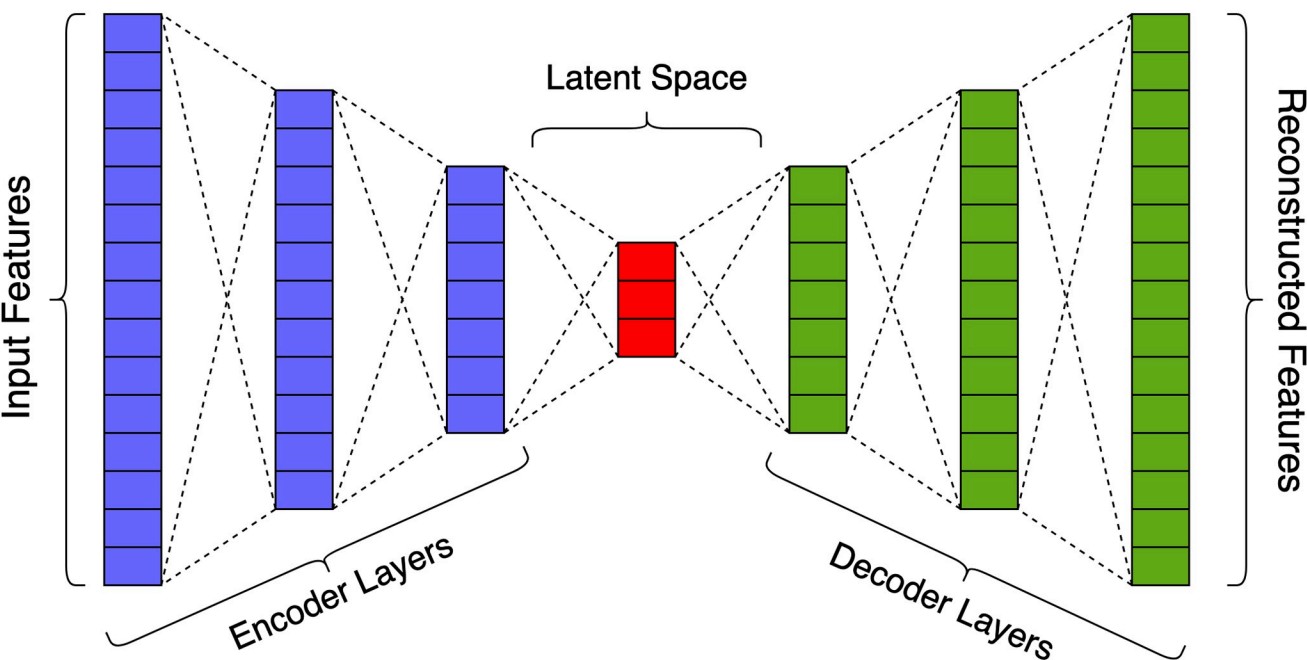

**Fig 1. Architecture of the AE network.** An AE comprises three components: the encoder, the latent space, and the decoder. Here, the encoder transforms the input data to an encoded representation, the latent space is a representation of the compressed knowledge of the input data, and the decoder reconstructs the input data from its encoded form.

where

$$AE(X) = D(Z) \quad Z = E(X) \tag{5}$$

and $\| \cdot \|_2$ denotes L2-norm.

The AE-based anomaly detection uses the reconstruction error as the anomaly score attempting to recreate the original data after a nonlinear compression process. Here, the network is trained using only normal data to learn all necessary characteristics and the relationships among its input features. During the model inferencing, the network reconstructs the normal sequences while failing to reconstruct the abnormal sequences. The AE network is used to detect anomalies on each local device. Training is conducted with the preprocessed data sequences and the pre-trained model is utilized to detect anomalies during inference.

## 4.4 Federated learning

In traditional distributed machine learning, central server data aggregation can pose issues like increased computational costs, and high network bandwidth, especially when dealing with IoT device data [9, 29]. To address these challenges, we propose the privacy-preserving FLAE network. It consists of two components: the server side, responsible for sharing and updating the global model, and the client side, comprising multiple devices with power consumption information. Our network enables training a global model while keeping the training dataset on local devices. In the following sections, we discuss the FL architecture and aggregating process in detail.

**4.4.1 Federated learning architecture.** Here, we describe the FL architecture. Assume that there are $K$ devices, each of which has a private dataset $\mathcal{D}_k$, where $k = 1, \ldots, K$. The data $\mathcal{D}_k$ of the $K^{th}$ devices are not shared with the server, in contrast to traditional distributed

learning processes that collect and use all local data $\mathcal{D} = \cup_{k=1}^{K} \mathcal{D}_k$ from all devices to train a model [21]. For every device involved in the training process (also referred to as clients), the data preprocessing stage is exactly the same as that describe for the local training. The main building blocks of this proposed method are the local training, server aggregation, and global model broadcasting, as shown in Fig 2.

*Local training.* The device data consist of power consumption data recorded at regular intervals and are commonly affected by noise caused by faulty devices. To address this issue, the data is preprocessed by splitting it into training data, testing data, and labels. Labels are manually added by doubling the testing data value and designating them as anomalies. The preprocessed training data is then used to train an autoencoder (AE) model, which is subsequently used to evaluate the unseen test data. The reconstruction loss is used to adjust the threshold value, where any value above the threshold is considered an anomaly. Each device then sends its pre-trained model weights to the server for aggregation, as illustrated in Fig 3.

*Server aggregation.* The server aggregator represents a robust cloud server with rich computing resources. It serves two fundamental functions in the context of distributed machine learning, namely (1) initializing and distributing the global model to all participating devices, and (2) collecting the model weights contributed by the client devices. The training process between the server and the clients takes place in rounds, during which the server aggregates the local model weights using the Federated Averaging (FedAvg) algorithm. This iterative approach enables the global model to be updated while preserving the privacy and security of the clients' data [13]. The training process is repeated until the global model reaches a state of convergence, indicating that it has achieved optimal performance.

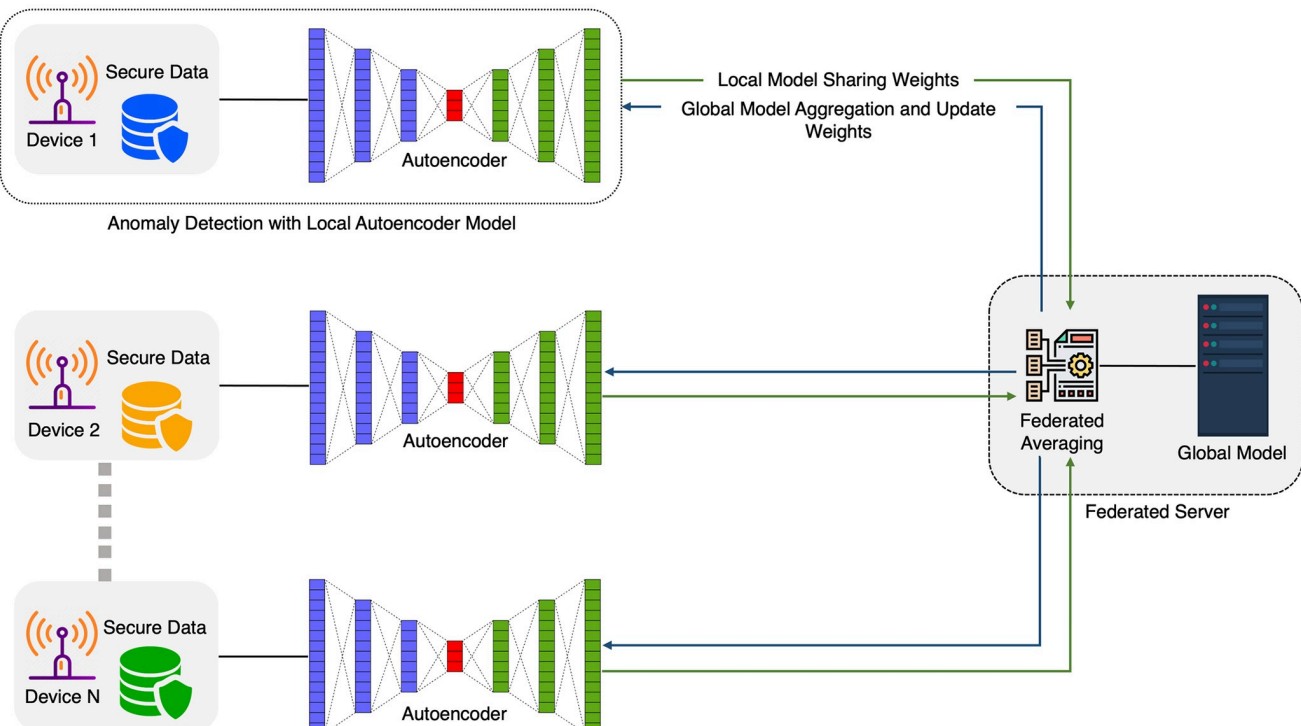

**Fig 2. Architecture of the anomaly detection model using the proposed FLAE.** When the FL process is initiated, all devices begin training simultaneously. Here, each device sends its weights to the global server, which then uses the FedAvg algorithm to aggregate the received weights. The global server then sends the aggregated weight back to each device and training continues.

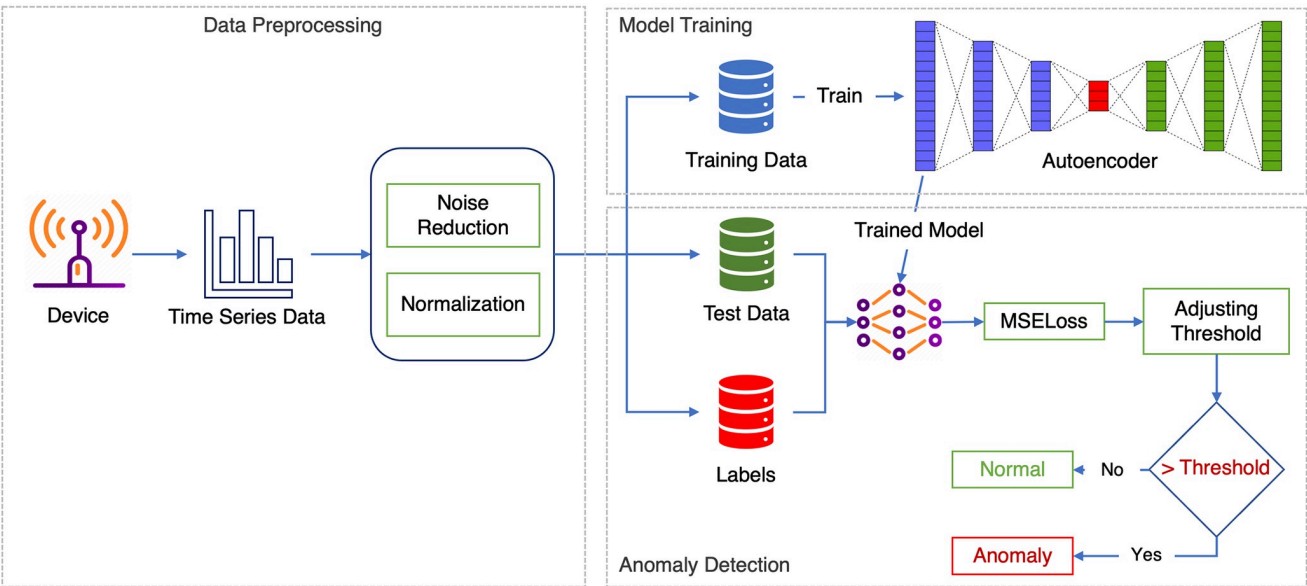

**Fig 3. Overview of the entire process of the local AE model, from data preprocessing to anomaly detection.** The dataset is split into training data, testing data, and label data. Label data are added manually by doubling their value. Once the model is trained, the MSELoss is calculated to define the threshold as the reconstruction error of the training data to detect anomalies. If the loss is greater than the threshold, the datapoint is labeled as an anomaly.

*Global model broadcasting.* Once the server has updated the model weights, it broadcasts the updated model to each device. This ensures that every device receives the latest global model and is able to use it for improved accuracy in anomaly detection. Each device replaces its local model's weights with the updated global weights to initiate a new round of training. This process is essential to keep all the devices in sync and maintain a consistent global model across the network.

**4.4.2 Federated averaging algorithm.** Federated optimization is built from stochastic gradient descent (FedSGD), which can be applied to the federated optimization problem, where a single batch gradient calculation is performed per round of communication [30]. Note that this technique is computationally efficient; however, it requires large numbers of training rounds to produce an effective model. As mentioned previously, FL first chooses $C$ fractions of the clients in each round in order to calculate the loss gradient over all data stored by the selected client devices. The goal is to minimize the global model loss function on all distributed datasets.

In a typical implementation of FedSGD with $C = 1$ and a fixed learning rate $\eta$, each client $k$ computes the following:

$$g_k = \nabla F_k(w_t) \tag{6}$$

where

$$F_k(w) = \frac{1}{n_k} \sum_{i \in \mathcal{D}_k} f_i(w) \tag{7}$$

where $w_t$ represents the model weights in communication round $t$, $\mathcal{D}_k$ is a set of data points on client $k$, and $n_k$ is the size of the dataset on client $k$. The average gradient on each client local data at the current model $w_t$, and then the central server aggregates these gradients and applies

the following update:

$$w_{t+1} \leftarrow w_t - \eta \sum_{k=1}^{K} \frac{n_k}{n} g_k \qquad (8)$$

Here, $\sum_{k=1}^{K} \frac{n_k}{n} g_k = \nabla f(w_t)$; thus, an equivalent update is given by $\forall k, w_{t+1}^k \leftarrow w_t - \eta g_k$:

$$w_{t+1} \leftarrow \sum_{k=1}^{K} \frac{n_k}{n} w_{t+1}^k \qquad (9)$$

where $w_{t+1}^k$ is the model weights in communication round $t$ on client $k$. Each client performs one step of gradient descent on its own local data. The server then takes the weighted average of the resulting models. As a result, we can add more computation to each client by iterating the local update multiple times prior to performing the averaging step:

$$w^k \leftarrow w^k - \eta \nabla F_k(w^k) \qquad (10)$$

This minor modification results in the FedAvg algorithm. Here, there are three parameters that control the amount of computation, i.e., $C$, (the percentage of clients that calculate on each round), $E$ (the number of training iterations each client runs on its local dataset in a round) and $B$ (the size of the local minibatch used for client updates).

## 4.5 Anomaly detection

**4.5.1 Anomaly score.** In the context of multivariate time-series data analysis, we calculate the anomaly score for each observation by employing a comprehensive approach. Specifically, the anomaly score is defined as a measure of deviation from the expected behavior. The anomaly score is determined by the following equation.

$$s = \frac{1}{N} \sum_{i=1}^{N} \|X - AE(X)\|_2 \qquad (11)$$

where the loss function, $\|X - AE(X)\|_2$ is the amount of error in the reconstruction of the AE in the compression network. The reconstruction probability is used as the anomaly score in our model. A low score means the input $X$ can be well reconstructed. A higher score indicates that the input is more likely to be anomalous. The next subsection describes how to automatically select the anomaly threshold value.

**4.5.2 Threshold selection strategy.** As is common in previous studies [27, 31–33], our study defines the anomaly threshold following the peak over threshold (POT) principle to select the threshold automatically and dynamically [34]. The basic concept of POT is to fit the data distribution by a generalized Pareto distribution and identify the appropriate extreme value to determine threshold values dynamically. Once we have the anomaly scores for a given timestamp for each dimension $s_i$, we label the timestamp as anomalous if this score is greater than the threshold.

## 5 Evaluation

### 5.1 Experimental setup

We provide a thorough performance analysis of the proposed FLAE method using power consumption datasets. We start by describing the evaluation metrics employed to assess the effectiveness of our method. Additionally, we discuss in detail the selection of experimental hyperparameters for our model. By conducting a rigorous experimental setup, we demonstrate

that our method achieves accuracy levels that are comparable to state-of-the-art methods, validating its effectiveness.

**5.1.1 Dataset.** We conducted a thorough analysis using a publicly accessible dataset of household power consumption, which can be found at https://doi.org/10.24432/C58K54 [35]. This dataset comprises multivariate time-series data that chronicles the patterns of electricity usage within a single household over a four-year period. In order to enhance the data's manageability, we converted the raw dataset into an hourly format. This modification substantially decreased the quantity of records, with each hourly entry encapsulating an aggregation of the original data for that specific hour. To simulate a distributed power system environment, we partitioned the transformed dataset into subsets and distributed them among six clients. Each client received approximately 6,000 records and acted as a power system emulator, actively participating in the collaborative training process. Following the collaborative training phase, we evaluated the resulting global model by testing it on the entire dataset, which encompasses all 36,000 records. The primary aim of this evaluation was to gauge the accuracy of the FLAE method in identifying anomalies within the data on household power consumption.

**5.1.2 Evaluation metric.** We conducted a thorough evaluation of the proposed FLAE method's performance using widely used evaluation metrics, namely precision (P), recall (R), and F1-score (F1), and their respective formulas are presented below.

$$P = \frac{TP}{TP + FP} \quad R = \frac{TP}{TP + FN} \quad F1 = 2 \cdot \frac{P \cdot R}{P + R} \tag{12}$$

where true positive, true negative, false positive, and false negative are denoted as TP, TN, FP, and FN, respectively. The number of appropriately detected anomalies is referred to as the TP, and TN is the number of cases in which the model correctly predicted a normal sequence. FN is the number of normal data that are inaccurately labeled as anomalies, and FP is the number of incorrectly predicted abnormal sequences as normal. We use commonly used metrics to evaluate the performance of all models. $P$ is the percentage of correct positive predictions to the total predicted positives, $R$ is the ratio of correct positive predictions to the total positives examples, and F1-score is a weighted average of precision and recall, respectively. Note that a higher F1-score value indicates better performance in terms of the model's ability to distinguish between anomalous and normal observations.

**5.1.3 Model hyper-parameter.** All experiments were conducted using the Python 3.7.13, PyTorch 1.12.0 [36], and Flower federated framework 0.19.0 environment on a Windows machine equipped with an Intel i7-12700F processor, 64GB of RAM, and an RTX 3060Ti graphics card. The AE network consisted of two main components: the encoder and decoder. The encoder network had four hidden layers, with dimension decreasing rates of 75%, 50%, 33%, and 25% respectively. The decoder had the same layer design as the encoder, but with an increasing sequence. We set the input dimension to be 8, which is the number of dataset features multiplied by the number of windows. The activation function inside the autoencoder was set to the hyperbolic tangent function (tanh). The model was trained using the Adam [37] optimizer with a learning rate of 0.0001 and a step scheduler size of 0.5 [38]. For this experiment, we split the dataset into 80% for training and 20% for testing, with a sliding window length of 15. We conducted 10 epochs of local model training, and a total of 20 training rounds were processed for the global model. The FL server and all six FL clients were simulated on the same machine to be trained in parallel, eliminating communication overhead. The performance result demonstrates the impact of hyperparameter tuning on the model.

## 5.2 Performance result

**5.2.1 Evaluation of noise reduction.** To evaluate the quality of the time-series data processed by the SG filter, we conducted a thorough evaluation by comparing the Mean Absolute Error (MAE), Mean Squared Error (MSE), and Root Mean Squared Error (RMSE) of data samples with and without the SG filtering. The SG filter was employed to compare the dataset between the state-of-the-art models and the proposed FLAE, owing to its remarkable effectiveness. The results presented in Table 1 demonstrate that the SG filter significantly improved the model's performance, resulting in a reduction of the loss function by 15% to 25%. This indicates that the model achieved a higher level of accuracy in predicting and detecting data abnormalities, as evidenced by the lower cost function.

**5.2.2 Power consumption prediction.** In this study, we evaluate the performance of the proposed FLAE model on the power consumption dataset. Our results, illustrated in Fig 4, demonstrate the high level of predictive accuracy achieved by the FLAE model. To visualize the performance of our predictive model, we plotted the actual and predicted values for a period of 30 days. The actual data is depicted using a purple line, whereas the red line represents the corresponding predicted values. The forecast data generated by our proposed FLAE model closely aligns with the actual test data, as evidenced by the visual comparison.

**5.2.3 Anomaly detection.** We evaluated the effectiveness of the proposed FLAE approach using a methodology similar to that employed in [20], wherein we manually inserted anomaly points into the test dataset. We evaluate the anomaly detection of the proposed FLAE model through comparison with state-of-the-art models. The chosen model includes AE, DAGMM [39] OmniAnomaly [33], MAD_GAN [40], and USAD [41]. The model's performance is evaluated using accuracy, precision, recall, AUC, and F1-score values [42, 43]. Table 2 shows a comparison of the accuracy, precision, recall, AUC, and F1-score metrics obtained by the state-of-the-art models and the proposed FLAE model on the power consumption dataset. The F1-score, which represents the balance between precision and recall, is a crucial indicator of a model's effectiveness. Our proposed FLAE model achieves comparable performance in detecting power consumption abnormalities to other state-of-the-art models.

Additionally, we used POT to automatically and dynamically generate thresholds from the AE model reconstruction error to evaluate the test dataset. Data points with scores greater than the threshold are considered anomalous. The total data length is approximately 7000 data points plotted against the threshold value in the test data in Fig 5. The green line represents the anomaly score, the red line indicates the threshold value, and the red dots represent abnormal values.

**5.2.4 Federated VS. centralized.** The non-FL models depend on a centralized approach, gathering training data on a central server. Although this method simplifies the training process, it may result in increased data transfer bandwidth. In contrast, the FL technique in the

**Table 1. MAE, MSE, and RMSE values of SG filter on the power consumption datasets.**

|  | MAE | | MSE | | RMSE | |
| --- | --- | --- | --- | --- | --- | --- |
|  | Original | Filter | Original | Filter | Original | Filter |
| AE | 0.039 | 0.036 | 0.003 | 0.002 | 0.054 | 0.044 |
| DAGMM | 0.042 | 0.036 | 0.004 | 0.002 | 0.063 | 0.044 |
| MAD_GAN | 0.030 | 0.022 | 0.002 | 0.001 | 0.044 | 0.031 |
| OmniAnomaly | 0.066 | 0.075 | 0.011 | 0.014 | 0.104 | 0.118 |
| USAD | 0.059 | 0.064 | 0.010 | 0.009 | 0.100 | 0.094 |
| **FLAE** | **0.050** | **0.042** | **0.004** | **0.003** | **0.063** | **0.054** |

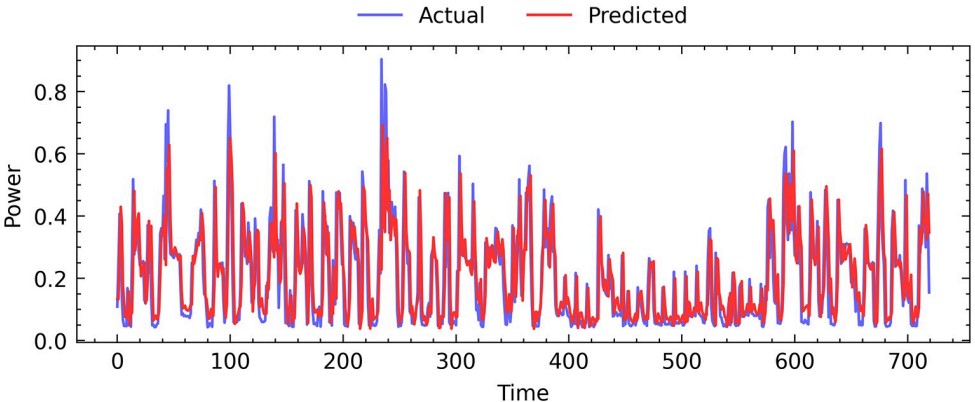

**Fig 4. The actual power consumption values are compared with the predicted values obtained through the proposed FLAE method.**

FLAE model allows collaborative device training with local data storage. FL improves computational performance and convergence, providing accurate results without data sharing. Fig 6 compares FLAE and non-FL models on F1 and AUC scores for six devices. FLAE performs well, accurately predicting anomalies in all devices. Non-FL models only detect anomalies in three to four devices, lacking generalization due to limited training data. FL approaches enable accurate anomaly detection across distributed devices. The proposed FLAE method learns device characteristics by aggregating weights of local models. However, FLAE may perform lower than non-FL models in some cases due to data quality, device participation, and FL algorithm effectiveness.

**5.2.5 Ablation study.** An ablation study was performed to evaluate the effectiveness of the proposed FLAE and non-FL models under different configurations. The F1-score, AUC, and training time were evaluated for different window sizes and dataset training ratios. The missing points in the graphs comparison are very small values that make the graphs difficult to see.

**Training set size**. Fig 7 illustrates the variations in F1 score, AUC, and training time resulting from the application of the FLAE method on the dataset. Different training data ratios, ranging from 20% to 80%, were considered. An analysis of the results shows that as the training data size increases, the prediction performance improves, but the training time also increases. Upon analyzing, it was determined that utilizing 60% to 80% of the dataset for training optimizes a balance between maintaining the training time and achieving high model accuracy. It is important to note that the proposed FLAE model was not included in the ablation study since the FL model does not have access to centralized data.

**Table 2. Performance comparison of proposed FLAE with the state-of-the-art models on the power consumption dataset.**

|  | Power Consumption Data | | | | |
| --- | --- | --- | --- | --- | --- |
|  | Accuracy | Precision | Recall | AUC | F1 |
| AE | 0.9968 | 0.9896 | 0.9995 | 0.9976 | 0.9945 |
| DAGMM | 0.9973 | 0.9911 | 0.9995 | 0.9979 | 0.9952 |
| MAD_GAN | 0.9231 | 0.9959 | 0.7380 | 0.8684 | 0.8478 |
| OmniAnomaly | 0.9212 | 0.9873 | 0.7380 | 0.8670 | 0.8446 |
| USAD | 0.9970 | 0.9901 | 0.9995 | 0.9977 | 0.9947 |
| **FLAE** | **0.9968** | **0.9906** | **0.9995** | **0.9978** | **0.9950** |

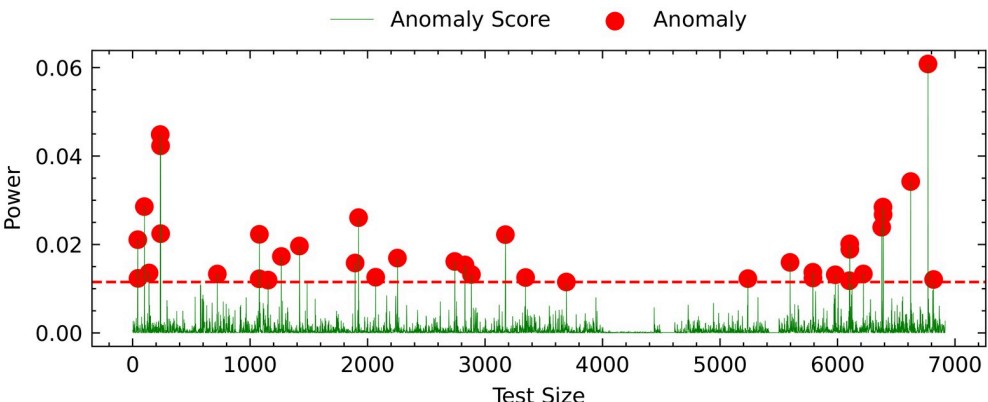

**Fig 5. Anomaly score of the active power and anomalies exceeding the threshold for 7000 hours.** The green line represents the anomaly score, calculated by measuring the deviation between actual and predicted values. The red dots on the graph indicate anomalies that exceed the threshold value.

**Window size**. We investigated the performance of the proposed by adjusting the dataset's sliding window size, as shown in Fig 8. The window size affects both the anomaly scores and training times. For example, if the window is too small, then the local contextual information is not represented effectively. In contrast, anomalies that are short in duration can go undetected if the window size is too large. An enlargement in window size correlates with an increase in training time. After evaluating the performance, it was determined that a window

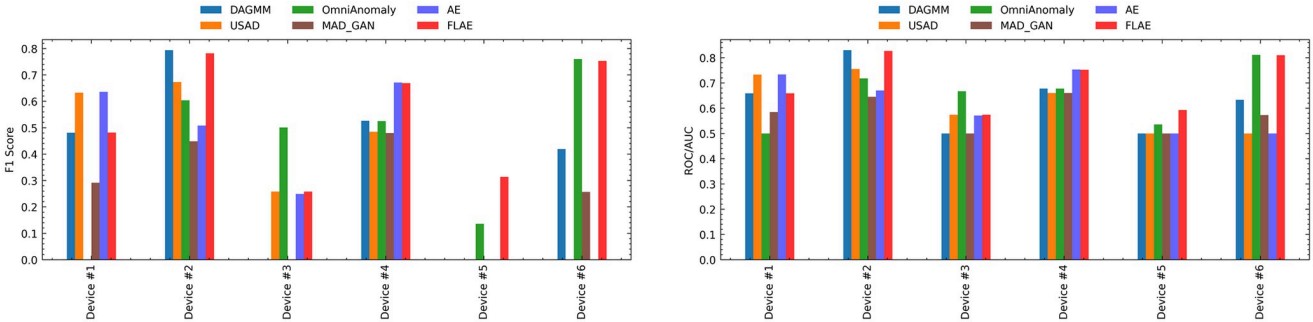

**Fig 6. Comparison between FL and non-FL models across six devices.** Results indicate that the proposed FLAE model performed well across all six devices, while non-FL models performed ineffectively in Device 5. This demonstrates the effectiveness of the proposed FLAE model in enabling accurate anomaly detection across a distributed network of devices.

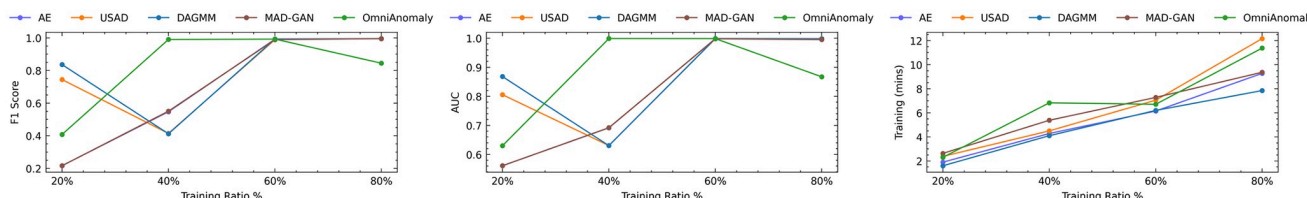

**Fig 7. The performance of the non-FL models trained on the power consumption dataset are evaluated for different dataset training sizes.** The F1-score and AUC score, as well as the training time, are reported for training sizes ranging from 20% to 100% of the full dataset.

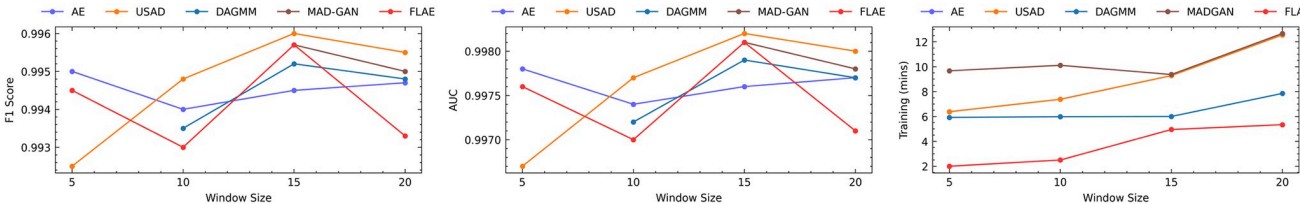

**Fig 8. The performance metrics of F1-score and AUC score, as well as the training time, were evaluated for the power consumption dataset using different window sizes ranging from 5 to 20 of the sliding window.**

size of 15 offers a reasonable balance between F1-score, AUC score, and training time. It is important to note that the OmniAnomaly model was not included in the comparison graph as it does not employ the sliding window technique.

## 6 Conclusion

With the growing demand for electricity resulting from population growth and technological advancement, it has become increasingly crucial to accurately predict electric power consumption and detect anomalies while ensuring efficient power distribution systems. In this paper, we have proposed the FLAE method to detect anomalies in a power system's time-series data without needing to share data. Our proposed method utilizes the power of federated learning, enabling multiple entities to collaboratively train a model on their respective local datasets. The proposed FLAE method is comprised of a local and global model architecture. At the local level, an autoencoder model is utilized to capture an individual's IoT device data and extract relevant features. At the global level, a model aggregator is employed to collect and aggregate the local model weights, update the global model parameters, and share them with other devices across the system. The proposed FLAE prediction and anomaly detection performance was evaluated on a power consumption dataset with non-FL state-of-the-art techniques. The results indicate that the FLAE model achieved a high level of prediction accuracy and performed similarly to the non-FL state-of-the-art models. In addition, we conducted an ablation study of the proposed FLAE method by evaluating its performance and robustness under different configurations by varying the sliding window size, ranging from 5 to 20, and the dataset set ratio between 20% and 80%. The results of these experiments demonstrate that FLAE is capable of detecting abnormalities in datasets across various configuration setups. In our future research, we aim to conduct a comprehensive analysis of communication and computational power costs using real IoT devices to provide a more accurate representation of the system's performance. Additionally, we plan to delve deeper into the security of data, examining various threat models and proposing countermeasures to mitigate security risks.

## Author Contributions

**Conceptualization:** Kimleang Kea, Tae-Kyung Kim.

**Data curation:** Kimleang Kea.

**Formal analysis:** Kimleang Kea, Youngsun Han, Tae-Kyung Kim.

**Funding acquisition:** Youngsun Han.

**Investigation:** Kimleang Kea, Youngsun Han, Tae-Kyung Kim.

**Methodology:** Kimleang Kea, Tae-Kyung Kim.

**Project administration:** Youngsun Han, Tae-Kyung Kim.

**Resources:** Kimleang Kea.

**Software:** Kimleang Kea.

**Supervision:** Tae-Kyung Kim.

**Validation:** Kimleang Kea, Tae-Kyung Kim.

**Visualization:** Kimleang Kea.

**Writing – original draft:** Kimleang Kea.

**Writing – review & editing:** Kimleang Kea, Youngsun Han, Tae-Kyung Kim.

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
