## [Decision Letter · Decision Letter 0]

9 Jun 2023

PONE-D-23-09375Enhancing Anomaly Detection in Distributed Power Systems using Autoencoder-based Federated LearningPLOS ONE

Dear Dr. Kim,

Thank you for submitting your manuscript to PLOS ONE. After careful consideration, we feel that it has merit but does not fully meet PLOS ONE’s publication criteria as it currently stands.Therefore, we invite you to submit a revised version of the manuscript that addresses the points raised during the review process.

Please take into consideration all the reviewers' comments while preparing the revised version of your manuscript. Moreover, be sure to provide more details of your dataset and to make it publicly available for the sake of reproducibility of the study.

We look forward to receiving your revised manuscript.

Kind regards,

Letterio Galletta

Academic Editor

PLOS ONE

Journal Requirements:

“This work was conducted under the framework of the research and development program of the Korea Institute of Energy Research (C1-2418).”            

“This work was conducted under the framework of the research and development 526 program of the Korea Institute of Energy Research (C1-2418)”

“This work was conducted under the framework of the research and development program of the Korea Institute of Energy Research (C1-2418).”

7. PLOS requires an ORCID iD for the corresponding author in Editorial Manager on papers submitted after December 6th, 2016. Please ensure that you have an ORCID iD and that it is validated in Editorial Manager. To do this, go to ‘Update my Information’ (in the upper left-hand corner of the main menu), and click on the Fetch/Validate link next to the ORCID field. This will take you to the ORCID site and allow you to create a new iD or authenticate a pre-existing iD in Editorial Manager. Please see the following video for instructions on linking an ORCID iD to your Editorial Manager account: https://www.youtube.com/watch?v=_xcclfuvtxQ

8. Please upload a new copy of Figures 6, 7, 8 and 9 as the detail is not clear. Please follow the link for more information: https://blogs.plos.org/plos/2019/06/looking-good-tips-for-creating-your-plos-figures-graphics/

Reviewers' comments:

Reviewer's Responses to Questions

**Comments to the Author**

1. Is the manuscript technically sound, and do the data support the conclusions?

Reviewer #1: Yes

Reviewer #2: Partly

2. Has the statistical analysis been performed appropriately and rigorously? 

Reviewer #1: Yes

Reviewer #2: Yes

3. Have the authors made all data underlying the findings in their manuscript fully available?

Reviewer #1: No

Reviewer #2: No

4. Is the manuscript presented in an intelligible fashion and written in standard English?

Reviewer #1: Yes

Reviewer #2: Yes

5. Review Comments to the Author

Reviewer #1: In this manuscript, the authors present a federated learning framework for anomaly detection in distributed power systems. With the exception of some grammatical errors, the paper is relatively well written and technically sound. My general comments may be summarized as follows:

* Please ensure that all acronyms that are used more than once are defined at first occurrence; those that are used only once need to be spelled out.

* For the sake of clarity, vectors and matrices should be denoted by boldface lower- and upper-case letters, respectively.

* English usage, grammar or spelling errors require some copy-editing.

* Line 12 should be rephrased as "{0,1} with 0 and 1 representing “normal” and “anomalous” data points, respectively."

* Eq. (3) is unnecessary and should be removed.

* Line 253: delete "or mean squared error".

* All figures should be included in the text (not at the end of the paper).

* Eq. (12): How is the loss function defined? what are O_i and \\hat{W}_i? and what is the difference between the inputs X and W?

* The labels in the figures are too small and virtually unreadable.

* The runtime analysis of the proposed algorithm needs to be discussed.

* It is unclear how the hyper-parameters affect the overall performance of the proposed approach.

Reviewer #2: Please provide more details of the dataset. Is it publicly available? "distributed power system" in the title is misleading as the paper actually uses a home energy consumption dataset. Please provide more justifications for the use of FL for this application.

6. PLOS authors have the option to publish the peer review history of their article (what does this mean?). If published, this will include your full peer review and any attached files.

Reviewer #1: No

Reviewer #2: No

---

## [Author Response · Author response to Decision Letter 0]

19 Jun 2023

To: PLOS ONE Editor

Re: Response to reviewers

We are writing to express our gratitude for the time and effort you have dedicated to reviewing our paper, "Enhancing Anomaly Detection in Distributed Power Systems using Autoencoder-based Federated Learning." We deeply appreciate the constructive feedback and suggestions you have provided, which have significantly contributed to the enhancement of our work.

We have thoroughly considered all your comments and suggestions and have made the requisite amendments to our manuscript accordingly. We believe that the revised paper effectively addresses all the issues brought up in your review, and we are hopeful that it aligns with your expectations.

Once again, we wish to express our gratitude for your time and consideration. We eagerly anticipate any further feedback you may provide.

Thank you for your continued engagement with our work.

Sincerely,

Kimleang Kea, Youngsun Han, and Tae-Kyung Kim

Reviewer#1 

Comments:

In this manuscript, the authors present a federated learning framework for anomaly detection in distributed power systems. With the exception of some grammatical errors, the paper is relatively well written and technically sound. My general comments may be summarized as follows:

• Please ensure that all acronyms that are used more than once are defined at first occurrence; those that are used only once need to be spelled out.

• For the sake of clarity, vectors and matrices should be denoted by boldface lower- and upper-case letters, respectively.

• English usage, grammar or spelling errors require some copy-editing.

• Line 12 should be rephrased as "{0,1} with 0 and 1 representing “normal” and “anomalous” data points, respectively."

• Eq. (3) is unnecessary and should be removed.

• Line 253: delete "or mean squared error".

• All figures should be included in the text (not at the end of the paper).

• Eq. (12): How is the loss function defined? what are O_i and \\hat{W}_i? and what is the difference between the inputs X and W?

• The labels in the figures are too small and virtually unreadable.

• The runtime analysis of the proposed algorithm needs to be discussed.

• It is unclear how the hyper-parameters affect the overall performance of the proposed approach.

Author response: We sincerely appreciate your thorough review of our manuscript, and we are grateful for your valuable comments and suggestions.

Author action: We have thoroughly deliberated on each of your observations and implemented the requisite modifications in response to them. The following are the amendments we have carried out in light of your feedback:

• We have taken care to define all acronyms at their first occurrence when they appear more than once in the text. For acronyms used only once, we have elected to spell them out fully to ensure clarity.

• To enhance readability and clarity, we have now denoted vectors and matrices by using boldface lower-case and upper-case letters, respectively.

• We have meticulously reviewed the manuscript for any errors related to English usage, grammar, and spelling to ensure the clarity and accuracy of our content.

• Upon re-evaluation of Line 12, we were unable to locate the specific reference mentioned in your comment.

• We have determined that Equation (3) was not essential to our discussion and have subsequently decided to remove it from the manuscript.

• We have excised the phrase "or mean squared error" from line 253 to mitigate any potential confusion.

• In an effort to enhance the narrative flow of the paper, we have now incorporated all figures directly within the text.

• Upon reexamination, we acknowledge that Equation (12) may have been misinterpreted due to its presentation in the initial version of our manuscript. To rectify this, we have revised Equation (12) to ensure its coherence with the rest of the equations in our manuscript.

• We have rectified the issue regarding the small and unreadable labels in the figures; we have ensured that all labels are now clearly visible for ease of understanding.

• The computational efficiency of the proposed algorithm is now demonstrated through a runtime analysis, as presented in the Ablation Study section, and is further expounded upon in our discussion.

• We have now included a comprehensive discussion on the impact of hyperparameters on the overall performance of our proposed approach, aiming to provide a thorough understanding of the model's behavior.

Reviewer#2

Comments:

Please provide more details of the dataset. Is it publicly available? "distributed power system" in the title is misleading as the paper actually uses a home energy consumption dataset. Please provide more justifications for the use of FL for this application.

Reviewer#2, Concern # 1: Please provide more details of the dataset. Is it publicly available?

Author response: We appreciate your inquiry concerning the dataset. In response to your query, we would like to confirm that the dataset utilized in this study is indeed publicly accessible. The dataset can be accessible from the following address: 

https://archive.ics.uci.edu/dataset/235/individual+household+electric+power+consumption

Author action: In response to your query, we have updated the manuscript to include appropriate citations for the datasets used in our study. Our intention is to provide a more transparent account to the readers regarding the sources of our datasets.

Reviewer#2, Concern # 2: "distributed power system" in the title is misleading as the paper actually uses a home energy consumption dataset. Please provide more justifications for the use of FL for this application.

Author response: We truly value your constructive comments and insights regarding our study. Indeed, we employ a home energy consumption dataset for our analyses. However, within the context of the Federated Learning (FL) framework, we use this dataset to emulate a "distributed power system" scenario. During this simulation, we partition the dataset into separate segments and allocate these to individual FL clients, each of which then operates as an autonomous power system. Each FL client serves to locally train models and subsequently consolidate their knowledge to the FL server in distributed manner.

Author action: We have taken your feedback into thorough consideration and have made substantial amendments to provide clarity on the usage of datasets in the context of the distributed power system, specifically in Section 5.1.1 of our manuscript. In the updated version, we have elucidated the relevant details to dispel any potential misunderstanding related to this matter. We are confident that these revisions successfully address your concerns and offer a lucid understanding of the application of datasets in our study.

---

## [Decision Letter · Decision Letter 1]

4 Aug 2023

Enhancing Anomaly Detection in Distributed Power Systems using Autoencoder-based Federated Learning

PONE-D-23-09375R1

Dear Dr. Kim,

We’re pleased to inform you that your manuscript has been judged scientifically suitable for publication and will be formally accepted for publication once it meets all outstanding technical requirements.

Kind regards,

Letterio Galletta

Academic Editor

PLOS ONE

Additional Editor Comments (optional):

Reviewers' comments:

Reviewer's Responses to Questions

**Comments to the Author**

1. If the authors have adequately addressed your comments raised in a previous round of review and you feel that this manuscript is now acceptable for publication, you may indicate that here to bypass the “Comments to the Author” section, enter your conflict of interest statement in the “Confidential to Editor” section, and submit your "Accept" recommendation.

Reviewer #1: All comments have been addressed

2. Is the manuscript technically sound, and do the data support the conclusions?

Reviewer #1: Yes

3. Has the statistical analysis been performed appropriately and rigorously? 

Reviewer #1: N/A

4. Have the authors made all data underlying the findings in their manuscript fully available?

Reviewer #1: Yes

5. Is the manuscript presented in an intelligible fashion and written in standard English?

Reviewer #1: Yes

6. Review Comments to the Author

Reviewer #1: Most of the points raised in my first review have been satisfactorily addressed in the revised manuscript.

7. PLOS authors have the option to publish the peer review history of their article (what does this mean?). If published, this will include your full peer review and any attached files.

Reviewer #1: No

---

## [Editor Report · Acceptance letter]

9 Aug 2023

PONE-D-23-09375R1 

Enhancing Anomaly Detection in Distributed Power Systems using Autoencoder-based Federated Learning 

Dear Dr. Kim:

I'm pleased to inform you that your manuscript has been deemed suitable for publication in PLOS ONE. Congratulations! Your manuscript is now with our production department. 

Kind regards, 

on behalf of

Dr. Letterio Galletta 

Academic Editor

PLOS ONE